# Challenges and Lessons Learned from a Field Trial on the Understanding of the Porcine Respiratory Disease Complex

**DOI:** 10.3390/vaccines13070740

**Published:** 2025-07-09

**Authors:** Elisa Crisci, Andrew R. Kick, Lizette M. Cortes, John J. Byrne, Amanda F. Amaral, Kim Love, Hao Tong, Jianqiang Zhang, Phillip C. Gauger, Jeremy S. Pittman, Tobias Käser

**Affiliations:** 1Department of Population Health and Pathobiology, College of Veterinary Medicine, North Carolina State University, Raleigh, NC 27607, USA; lmlorenz@ncsu.edu (L.M.C.); jjbyrne@ncsu.edu (J.J.B.); amandafigamaral@gmail.com (A.F.A.); 2Department of Chemistry & Life Science, United States Military Academy, West Point, NY 10996, USA; andrew.kick@westpoint.edu; 3K. R. Love Quantitative Consulting and Collaboration, Athens, GA 30608, USA; kim@krloveqcc.com; 4Department of Veterinary Diagnostic and Production Animal Medicine, College of Veterinary Medicine, Iowa State University, Ames, IA 50011, USA; haotong@iastate.edu (H.T.); jqzhang@iastate.edu (J.Z.); pcgauger@iastate.edu (P.C.G.); 5Smithfield Foods, Smithfield, VA 23430, USA; jpittman@smithfield.com; 6Department of Biological Sciences and Pathobiology, Center for Pathobiology, Immunology, University of Veterinary Medicine Vienna, 1210 Vienna, Austria; tobias.kaeser@vetmeduni.ac.at

**Keywords:** PRDC, PRRSV-2, immunity, NanoString

## Abstract

Background/Objectives: The porcine respiratory disease complex (PRDC) is a multifaceted, polymicrobial syndrome resulting from a combination of environmental stressors, primary infections (e.g., PRRSV) and secondary infectious agents (viruses and bacteria). PRDC causes severe lung pathology, leading to reduced performance, increased mortality rates, and higher production costs in the global pig industry. Our goal was to conduct a comprehensive study correlating both the anti-PRRSV immune response and 21 secondary infectious agents with PRDC severity. Methods: To this end, PRRSV-negative weaners were vaccinated with a PRRSV-2 MLV and put into a farm with a history of PRDC. Subsequently, anti-PRRSV cellular and antibody responses were monitored pre-vaccination, at 28 days post vaccination (dpv) and during PRDC outbreak (49 dpv). NanoString was used to quantify 21 pathogens within the bronchoalveolar lavage (BAL) at the time of necropsy (51 dpv). PRRSV-2 was present in 53 out of 55 pigs, and the other five pathogens (PCMV, PPIV, *B. bronchiseptica*, *G. parasuis*, and *M. hyorhinis*) were detected in BAL samples. Results: Although the uncontrolled settings of field trials complicated data interpretation, multivariate correlation analyses highlighted valuable lessons: (i) high weaning weight predicted animal resilience to disease and high weight gains correlated with the control of the PRRSV-2 field strain; (ii) most pigs cleared MLV strain within 7 weeks, and the field PRRSV-2 strain was the most prevalent lung pathogen during PRDC; (iii) all pigs developed a systemic PRRSV IgG antibody response which correlated with IgG and IgA levels in BAL; (iv) the induction of anti-field strain-neutralizing antibodies by MLV PRRSV-2 vaccination was both late and limited; (v) cellular immune responses were variable but included strong systemic IFN-γ production against the PRRSV-2 field strain; (vi) the most detected lung pathogens correlated with PRRSV-2 viremia or lung loads; (vii) within the six detected pathogens, two viruses, PRRSV-2 and PCMV, significantly correlated with the severity of the clinical outcome. Conclusions: While a simple and conclusive answer to the multifaceted nature of PRDC remains elusive, the key lessons derived from this unique study provide a valuable framework for future research on porcine respiratory diseases.

## 1. Introduction

Respiratory diseases in pigs are one of the most important health concerns for swine producers and the leading cause of mortality in nursery and grower–finisher units [1]. The term porcine respiratory disease complex (PRDC) was initially used to describe pneumonia of multiple etiologies that caused clinical disease with negative consequences on productive parameters during the finishing process. To date, the term delineates a multifaceted polymicrobial syndrome that results from a combination of infectious agents, environmental stressors, population size, management strategies, age, genetics, microbiome disruption [2,3] and impaired immunity; as a result, PRDC reduces pig performance and increases both mortality rates and production costs in the pig industry worldwide [1,4,5,6]. This multifactorial disease can affect ~30–70% of pigs upon break-out on a unit. Outbreaks usually occur at 8–20 weeks of age [1], often referred to as the “18-week wall” [7]. Morbidity rates associated with PRDC may range from 30 to 70% and mortality rates between 4% and 15% in affected farms [8,9].

PRDC is often the result of a combination of primary and secondary (or opportunistic) infectious agents: the primary pathogens subvert normal barriers and defense mechanisms to establish their own infection; then, the secondary pathogens take advantage of the virulent mechanisms of the primary agents to establish their infections [1,5,6]. Primary viral agents are porcine reproductive and respiratory syndrome virus (PRRSV), swine influenza A viruses (SwIAVs), pseudorabies virus (PRV), and porcine circovirus type 2 (PCV2); bacterial agents are *Mycoplasma hyopneumoniae*, *Bordetella bronchiseptica*, *Actinobacillus pleuropneumoniae*. Depending on the pathogens milieu some of the bacterial agents may act as both primary and opportunistic invaders. The most opportunistic ones are *Pasterella multocida*, *Streptococcus suis*, *Glaesserella* (*Haemophilus*) *parasuis*, *Trueperella* (*Arcanobacterium*) *pyogenes*, and *Salmonella* spp. [1,6]. Additionally, other minor viral pathogens have been associated with respiratory implications, e.g., viruses from Paramyxoviridae family (such as porcine rubulavirus, porcine parainfluenza virus (PPIV) and Nipah virus), porcine cytomegalovirus (PCMV), porcine respiratory coronavirus (PRCV), porcine parvoviruses (PPVs), porcine circovirus type 3 (PCV3) and porcine torque teno virus [5,6,10,11].

Infections with these multiple pathogens lead to lung lesions that are primarily located in the cranioventral parts of the lungs. They often present bronchopneumonia in combination with interstitial pneumonia and pleuropneumonia [12,13,14]. Therefore, histological lesions or histopathological diagnosis have often been used to evaluate associations between PRDC infectious agents and respiratory diseases worldwide [8,12,13,14,15,16].

Research across the globe has consistently identify PRRSV, PCV-2, SwIAV and *Mycoplasma* spp. as the most significant pathogens in respiratory diseases or associated with pneumonia-like gross lung lesions in slaughter pigs [8,15,16,17,18,19], with 88.2% of the cases caused by two or more pathogens [8]. Conversely, specific prevalence, lesion scores and coinfection patterns of pathogens can vary based on geographical regions [11,20] and swine production stage [14,21]. A Danish study detected *M. hyopneumoniae*, *M. hyorhinis*, and *P. multocida* most frequently among PRDC affected swine with a significant association found between lesion duration and a multifactorial infection: compared to subacute or chronic cases, coinfection with *P. multocida*, PCMV, PCV2, *M. hyopneumoniae*, and *M. hyorhinis* was seen more frequently in acute cases [12]. Ruggeri et al. [14] correlated pulmonary, pleural, and nasal lesions with the presence of pathogens and production stage: Compared to the growing group, the post-weaning group (i) showed more *S. suis*, PRRSV, *M. hyorhinis*, and *G. parasuis*, and (ii) presented catarrhal bronchopneumonia, pericarditis, and pleuro-pericarditis. In contrast, growing pigs were more likely to show *P. multocida*, PCV2, *M. hyopneumoniae*, and *A. pleuropneumoniae*, which was also largely associated with pleuropneumonia in fattening pigs [14]. In China, PRRSV, PCV2, and *G. parasuis* were most frequent in nursery pigs; in contrast, growing pigs had the highest detection rates of *P. multocida* and *A. pleuropneumoniae* [21].

The intricate interplay of pathogens in PRDC makes their individual ranking in disease development challenging. However, the pivotal roles of PRRSV, PCV2, and *M. hyopneumoniae* have led to a review evaluating their interplays and respective vaccines, with three key interactions: (i) PRRSV and Mycoplasma vaccinations can exacerbate PCV2 viremia; (ii) Mycoplasma vaccination may diminish PRRSV viremia; and (iii) PCV2 vaccination appears to compromise the efficacy of PRRSV vaccines [22]. Despite these numerous studies on PRDC and its associated pathogens, immune responses have rarely been analyzed or correlated with protection. Only one recent study analyzed lymphocytes responses in pigs with PRDC at three production stages—weaning (4 weeks old), initiation (8 weeks old), and growing phase (12–14 weeks old) [23]. Coinfections with PRRSV, PCV2, and H1N1 SwIAV were detected at all stages, while *M. hyopneumoniae* appeared only during initiation and growing stages. Weight gains were mainly reduced during the growing stage. Immune profiling revealed stage-dependent differences in blood CD4 T cell numbers and expression levels of subset-specific transcriptomic factors and pro-inflammatory cytokines. In infected farms the expression of GATA3 and IL4 were upregulated mainly during initiation and growing phase, indicating a Th2-prone response. For the Th1 responses, findings were contradictory: while Tbet expression was increased during initiation and growing phase, IFN-γ expression was decreased or completely absent during all stages [23].

From these worldwide studies, three main conclusions can be drawn: (i) PRDC is indeed a multifactorial disease that can lead to severe disease, reduced performance, and increased mortality; (ii) PRRSV is a key player; and (iii) the role of immunity, especially post-PRRSV vaccination, remains unclear.

To address this, we evaluate the kinetics of cellular and humoral responses to PRRSV vaccination, and their correlation with coinfections and disease severity in a PRDC field trial. We evaluated systemic cellular responses and PRRSV-specific antibodies in serum and in bronchoalveolar lavage (BAL). PRRSV loads were quantified in nasal swabs and serum via RT-qPCR, while BAL samples were analyzed for multiple PRDC pathogens using NanoString technology. Clinical signs, rectal temperatures and weight gains were monitored. At necropsy, gross lung lesions and histopathology were evaluated. In addition to study these parameters over time and on their own, both univariate and multivariate correlation analyses were performed. This integrated approach underscores the value of NanoString technology in mapping respiratory pathogen networks and provide novel perspectives on the intricate host–pathogen interactions in PRDC.

## 2. Materials and Methods

### 2.1. Study Design, Necropsy Procedure, Samples Collection and Processing

The study design is summarized in Figure 1. Sixty commercial Yorkshire 3-week-old piglets balanced by sex with an average weight of 6.4 kg (range 4.7–8.5 kg) were selected from a PRRSV-negative herd and introduced into a farm in North Carolina with history of PRDC. Upon arrival, animals were vaccinated with 2 mL of Ingelvac^®^ PRRSV-2 MLV vaccine (Boehringer Ingelheim, Ridgefield, CT, USA) intramuscularly at 4 weeks of age. Weight, rectal temperature, serum, whole-blood, and nasal swabs were collected prior to vaccination (0 days post vaccination, dpv), at 28 dpv and when PRDC was expected (49 dpv). At 51 dpv, animals were humanely euthanized using carbon dioxide gas inhalation followed by exsanguination. Then, lungs were harvested, and dorsal and ventral photos were taken; lungs were flushed with 1× PBS to collect BAL, and a total of five lung samples were taken for histology: one each from the right and left caudal lobes, right and left cranio-ventral lobes and then one from the accessory lobe. Blood sample processing was conducted as previously described [24]. Briefly, for serum collection, the blood was collected into SST tubes (BD Biosciences, San Jose, CA, USA), spun at 2000× *g* for 20 min at RT and stored in aliquots at −80 °C. The blood for peripheral blood mononuclear cell (PBMC) isolation was collected in heparin tubes (BD Biosciences, Franklin Lakes, NJ, USA) and performed by density centrifugation using Ficoll–Paque (GE Healthcare, Uppsala, Sweden). After isolation, PBMCs were frozen in 10% DMSO (Fisher Scientific, Waltham, MA, USA), 40% FBS (Biowest, Lakewood Ranch, FL, USA), and 50% RPMI-1640 with L-glutamine (Corning Inc., Corning, NY, USA) and stored in liquid nitrogen until they were thawed for laboratory analysis. The BAL was performed as previously described [25] and samples were stored at −80 °C. Partial sampling was performed on five pigs that succumbed during the experimental process.

### 2.2. Clinical Score, Lung Gross Lesions and Histopathology Scoring

Clinical scoring (or illness score) for determination of animal health status was performed at the end of the trial by personnel trained in the Individual Pig Care (IPC, Zoetis, Parsippany, NJ, USA) method [26]. IPC scores are generally calculated using an ABC scoring system with A referring to mild acute disease, B referring to moderate clinical disease and C referring to severe clinical disease. For this study, the ABC score was converted to a numeric score and was based upon overall assessment of the animal based on size, coughing and fitness, with 0 referring to healthy and 4 to severe clinical disease.

Necropsy samples were fixed in 10% Formaldehyde/Zn fixative (Electron Microscopy Sciences, Hatfield, PA, USA) for 24 h, transferred to 70% ethanol and stained with hematoxylin and eosin (H&E). Stained slides (histology) and lung photos (gross pathology) were forwarded to the ISU VDL and evaluated by a veterinary diagnostic pathologist who was blinded to treatments and necropsy dates. Macroscopic lung lesions and scores for interstitial pneumonia were performed as previously described [27]. Dorsal and ventral views of gross lung photos were evaluated for the percentage of the lung affected with pneumonia. Specifically, the percentage of pneumonia was estimated visually for each lung lobe, and the total percentage for the entire lung was calculated based on the weighted proportion of each lobe relative to the total lung volume. Each of the four cranioventral lobes represented 10%, the accessory lobe 5%, and each of the two caudal lobes represented 27.5% of the total lung (Appendix A).

Lung histopathology slides were blindly examined for microscopic lesions and each section, representing 7 different lobes of lung, were given a score based on the severity of interstitial pneumonia as follows: 0 = no microscopic lesions; 1 = minimal interstitial pneumonia; 2 = mild interstitial pneumonia; 3 = moderate interstitial pneumonia; 4 = abundant interstitial pneumonia; 5 = severe, multifocal interstitial pneumonia; 6 = severe, diffuse interstitial pneumonia (Appendix A). The interstitial pneumonia lesion scores were averaged for each pig for statistical analysis [27,28].

### 2.3. PRRSV-2 Isolation, Propagation, and Titration

The highly virulent PRRSV-2 field strain (referred as NC20-1, Lineage 1A, RLFP 1-4-4, GenBank accession number: OR805486.1) was initially isolated at South Dakota State University Animal Research and Diagnostic Laboratory (SDSU ARDL) and subsequently propagated at NCSU Veterinary College in MA104 cells, concentrated in a Sorvall 100S ultracentrifuge (Thermo Fisher Scientific, Newtown, CT, USA) at 73,000× *g* at 4 °C for 2 h, and titrated in MA104 cells as previously described [24,29]. During PRDC outbreak the PRRSV-2 field strain was verified with Open Reading Frame (ORF) 5 sequencing and restriction fragment length polymorphism (RFLP) by ISU VDL. An additional ORF5 and whole-genome sequencing (Cambridge Technologies, Tullamarine, VIC, Australia) was performed after propagation in MA104. NC20-1 sequence presents 84.2% whole-genome nucleotide homology, and 87.2% (GP5), 86.5% (GP4), 83.1% (GP3), 85.9% (GP2) amino acid homology with the PRRSV-2 MLV strain (GenBank accession number: AF066183.4). Evolutionary analyses were conducted in MEGA11 [30].

### 2.4. Viremia and Viral Shedding

Sera were shipped to Dr. Zhang’s laboratory at ISU VDL for RT-qPCR analysis of viremia using both PRRSV screening PCR and Ingelvac PRRS MLV vaccine-like PCR as previously described [31]. Briefly, nucleic acids were extracted from serum samples (100 µL), using a MagMAX™ Pathogen RNA/DNA Kit (Thermo Fisher Scientific, Waltham, MA, USA) and a Kingfisher Flex instrument (Thermo Fisher Scientific), and eluted into 90 µL of elution buffer. The nucleic acid extracts were then tested using a commercial real-time PRRSV screening RT-PCR kit (VetMAXTM PRRSV NA&EU Reagent v2, Thermo Fisher Scientific), which targets the conserved genomic regions and is capable of detecting and differentiating PRRSV-1 and PRRSV-2 strains. Additionally, a PRRSV-2 Ingelvac PRRS MLV vaccine-like real-time RT-PCR, which targets the nsp2 region, was used to specifically detect the presence of the Ingelvac PRRS MLV vaccine strain in the samples. Data analysis was conducted using Design & Analysis software version 2.7 (Thermo Fisher Scientific) with automatic baseline setting. The threshold was set at 0.05 for the PRRSV-1 detector of screening PCR and 0.1 for both screening PCR PRRSV-2 detector and Ingelvac MLV-specific PCR. A Ct cutoff of 37 was used for both PCR assays with Ct ≥ 37 reported as negative.

RNA from nasal swabs was extracted using the NucleoSpin Virus (Macherey-Nagel, Allentown, PA, USA), cDNA synthesis was performed using High Capacity cDNA RT kit (Applied Biosystem, Thermo Fisher Scientific, Waltham, MA, USA) and primers used for the detection of the genomic copies of PRRSV strains were nsp9 F (5′-CCTGCAATTGTCCGCTGGTTTG-3′) and nsp9 R (5′-GACGACAGGCCACCTCTCTTAG-3′), previously described by Spear and Faaberg [32].

### 2.5. Anti-PRRSV IgG and IgA Analyses

Serum and BAL fluids were tested with the PRRSV X3 ELISA (IDEXX, Westbrook, ME, USA) to determine total anti-PRRSV IgG presence. Additionally, total PRRSV IgA detection was performed in BAL by ISU VDL by modifying the IDEXX PRRS Oral Fluids Ab Test. Briefly, samples were diluted 1/200 with the sample diluent provided in the kit and incubated for 2 h at room temperature; an HRP-conjugate IgA (Bethyl Laboratories, Fortis Life Science, Boston, MA, USA) was used at 1/3000 dilution for 1 h at room temperature followed by substrate for 5 min at room temperature. In-house positive and negative controls replaced the kit controls.

### 2.6. Neutralizing Antibodies Against PRRSV-2 Field Strain (NC20-1)

Serum samples from pigs were shipped to SDSU ARDL where fluorescent focus neutralization (FFN) assay was conducted to measure neutralizing antibody (NA) titers against the NC20-1 field strain. The reciprocal of the highest serum dilution resulting in ≥90% reduction in the number of fluorescent focus forming units as compared to the negative serum control was defined as the neutralizing antibody titer [33].

### 2.7. ELISpot IFN-γ Assay

Washed PBMC of each animal were resuspended in RPMI-1640 supplemented with 10% FBS (Biowest, USA), 10,000 IU Penicillin, 10 mg/mL Streptomycin (Corning Inc., Corning, NY, USA), and 50 mg/L gentamicin (Thermo Fisher Scientific, Waltham, MA, USA). MultiScreenHTS IP, 0.45 µm, Filter Plate (MilliporeSigma, Burlington, MA, USA) were activated with 35% ethanol, washed, and coated overnight with 10 μg/mL of IFN-γ capture antibody (anti-porcine IFN-γ mAb pIFN γ-I) (Mabtech, Nacka Strand, Sweden) diluted in PBS, 100 μL/well. After removing the coating antibody with five PBS washings, 100 μL/well of PBMC were dispensed in coated wells at density of 5 × 10^5^ cells/well. Then, cell cultures were stimulated with 100 μL/well of PRRSV-2 field strain NC20-1 at MOI = 3, Concavalin A (ConA) (Thermo Fisher Scientific, Waltham, MA, USA) at 5 μg/mL or were mock-stimulated with culture medium. After overnight incubation at 37 °C in 5% CO_2_, cells were removed, plates were washed with PBS and 100 μL/well of biotinylated detection antibody (1 μg/mL, anti-IFN-γ clone P2C11, Mabtech) was added and incubated 1 h at 37 °C. After washing, reaction was revealed by sequential incubation of plates with streptavidin alkaline phosphatase (1:2000, Mabtech) for 1 h, 37 °C, and 5-bromo-4-chloro-3-indolyl phosphate/nitro blue tetrazolium substrate (Sigma-Aldrich, St. Louis, MO, USA) for 25 min at room temperature. Spot development was blocked by washing plates under tap water and letting them air dry. Spots were counted on ASTOR ELISpot reader (Mabtech) and animals with more than 30 spots were arbitrarily considered responders. Frequencies of spot forming units (SFUs) were expressed as SFU per 500,000 of PBMC.

### 2.8. NanoString Gene Expression Analysis

Pathogen prevalence in BAL samples was evaluated using NanoString gene expression technology. BAL (400 µL) was diluted in Trizol (800 µL) for RNA extraction, RNA preps were diluted to a concentration of 20 ng/μL. Using a uniquely designed swine pathogens CodeSet (nCounter XT CodeSet Gene Expression Assay NanoString Technologies Seattle, WA, USA) (Appendix A), hybridization buffer was added to the reporter CodeSet to create a master mix. An aliquot of the master mix was mixed with 5 μL of each RNA sample, followed by the capture probe. The samples were placed at 65 °C in a pre-heated thermal cycler for at least 18 h. One sample per lane was loaded and then processed in the nCounter (NanoString Technologies). The custom-designed CodeSet was selected to identify the following swine pathogens: PRRSV (VR-2332 type, vaccine strain), PRRSV (universal, all strains), PCV2 PCV3, Nipah virus, PPV, SwIAV subtypes (H1, H3, N1, N2), suid herpesvirus 1 (SuHV-1, or pseudorabies virus (PrV) or Aujeszky’s disease virus), PRCV, Torque Teno Sus virus, porcine rubulavirus, PPIV, PCMV, *Actinobacillus suis*, *Actinobacillus pleuropneumoniae*, *Mycoplasma hyopneumoniae and hyorhinis*, *Pasteurella multocida*, *Trueperella pyogenes*, *Bordetella bronchiseptica*, *Glaesserella* (*Haemophilus*) *parasuis*. One housekeeping gene (ABCF1, *Sus Scrofa* gene) was included in the CodeSet. Gene copy numbers were normalized to 1 mL of BAL. The threshold was calculated based on presumed negative samples in the non-normalized analysis (value = 22) and then it was multiplied with the averaged normalization value (final value of threshold = 238). A portion of the BAL samples analyzed by NanoString were validated at ISU VDL by RT-qPCR. One negative BAL sample and six positive BAL samples (total *n* = 7) were tested by ISU VDL for specific swine pathogens (PRRSV-2, *G. Parasuis*, *M. hyorhinis*, PCMV, PPIV1 and SwIAV) via RT-qPCR (Table 1).

### 2.9. Statistical Analysis

Some basic statistical analyses were performed in GraphPad Prism 10 (Boston, MA, USA). More comprehensive statistical analyses were performed using R Statistical Software (v4.3.2; R Core Team 2024) [34]. Spearman correlations were initially performed for all pairs of numeric and ordinal variables in the data set, to provide basic information about the strength of individual relationships. Based on hypothesized relationships among the variables, three sets of analyses were performed: (i) those predicting PRRSV pathogen levels from immunity measures, (ii) those predicting levels of other pathogens from immunity measures and levels of PRRSV, and (iii) those predicting PRDC severity measures from a combination of immunity measures, levels of PRRSV, and levels of other pathogens. For all models, variance inflation factors (VIFs) were calculated for all predictors to determine whether high multicollinearity among independent variables would impede interpretation of the final model. Following, due to the large number of potential predictors for relative to the number of observations, regression analyses with a stepwise procedure relying on the Akaike information criterion (AIC) were run. These models iteratively add (or remove) independent variables according to the step that most improves (increases) the AIC; variable selection ceases when addition or removal of a variable no longer improves this criterion. The type of regression model used depended on the distribution of the outcome variable. For numeric/continuous outcomes, multiple linear regression models were used, sometimes with a transformation of the outcome to ensure normality and therefore appropriateness of the model. For the binary outcome of whether an individual pig tested positive for non-vaccine like PRRSV, logistic regression was used. For the ordinal outcome of sick score (an ordered category from 0 to 4), ordinal logistic regression was used.

## 3. Results

The overall goal was to determine the role of not only PRRSV infection and immunity but also other viral and bacterial respiratory pathogens in PRDC. To that end, we evaluated the overall health of pigs, PRRSV loads and anti-PRRSV immunity in blood, BAL, and nasal swabs, and used NanoString technology to quantify 21 respiratory pathogens in BAL. Of these 21 pathogens, the six most prevalent ones (*G. parasuis*, *M. hyorhinis*, *B. bronchiseptica*, PRRSV-2, PCMV, and PPIV) were included in an in-depth statistical analysis on the relations of PRRSV immunity, pathogen prevalences, and PRDC severity. All data are summarized in Appendix A and Spearman correlation analyses in Appendix A.

### 3.1. Weight Gains, Clinical Evaluation, and Lung Pathology

As shown in Figure 2A, weights increased over time from an average of 6.4 kg (0 dpv, 4 weeks of age) over 14.1 kg (28 dpv, 8 weeks of age) to 26.9 kg (49 dpv, 11 weeks of age). With four exceptions, weight gains also increased over time with 7.7 kg from 0 to 28 dpv and 12.8 kg from 28 to 49 dpv (Figure 2A).

Using Spearman Correlation analyses, weights on day 0 showed a large correlation (ρ = 0.571) with weights on day 28; these strongly correlated with weights on day 49 (ρ = 0.724). In addition, weights on day 28 (ρ = 0.884) and day 49 (ρ = 0.806) strongly correlated with weight gains over the respective periods. This emphasizes the importance of the pre-weaning period on the overall pig weight even long after weaning. Furthermore, weight gain from day 0 to 28 had a positive relation with lower rectal temperatures at day 49 (*p* = 0.009), and weight at 49 dpv inversely correlated with temperature (ρ = −0.445) at the same time point (Appendix A). Pig weights at both days 28 and 49 also strongly inversely correlated with sick score (Figure 2C, ρ = −0.562 and −0.601, respectively). This demonstrates that a “healthy” (high) pig weight promotes pig resilience to disease.

The average body temperature was 40.2 °C and 39.6 °C at 28 and 49 dpv, respectively. Of note, while nine pigs had fever (≥40.5 °C) at 28 dpv, only one pig had a fever during the presumed PRDC period (49 dpv, Figure 2B). Despite that lack of fever, most pigs often showed clinical signs of sickness (average: 1.0), and histological lung lesion (average: 2.5) scores as well as gross lung lesions affecting mostly 75–100% of the lung at time of necropsy (51 dpv, Figure 2C and Appendix A). Of note, the intersections of the grey lines connecting the same animals in this figure visualize that overall sickness scores (medical observation of the pig overall) were not able to predict histological (H&E) lung pathology. In addition, in nearly all animals and despite the absence of fever, most of the lungs showed pathological lesions during the presumed PRDC period (51 dpv).

### 3.2. Viremia, BAL Viral Loads, and Viral Shedding of PRRSV-2

All serum samples at 0, 28 and 49 dpv tested negative for PRRSV-1. At day 0 dvp, all the evaluated 56 pig serum samples tested negative by both PRRSV-2 screening PCR and Ingelvac MLV vaccine-like PCR (Appendix A). By 28 dpv, 55/56 pigs developed viremia, as evidenced by the positive results in both the PRRSV screening PCR (Figure 3A, Table 2) and the Ingelvac MLV vaccine-like PCR (Figure 3B, Table 2), indicating the presence of Ingelvac MLV vaccine virus in these samples. For samples containing only the Ingelvac PRRS MLV vaccine virus, the Ct value of the Ingelvac MLV vaccine-like PCR is typically higher, but within 3 Ct, compared to the Ct value of the PRRSV screening PCR [31]. In the current study, a Ct difference of ≥5 between the PRRSV screening PCR and the Ingelvac MLV vaccine-like PCR was used to infer coinfection of both the MLV virus and field strain NC20-1. Using this criterion, 21 animals were likely coinfected with both the field strain NC20-1 and the Ingelvac MLV virus (Table 2).

By 49 dpv, among the available 51 pig serum samples, PRRSV-2 MLV strain was cleared from blood in all but one animal (Figure 3B, Appendix A). In contrast, all but 2 animals (49/51) showed PRRSV-2 field strain viremia at 49 dpv (Figure 3A, Appendix A). These data show that while at 28 dpv 37.5% of the animals were probably coinfected with both the MLV-like and PRRSV-2 field NC20-1 strains, at 49 dpv, viremia was nearly solely caused by PRRSV-2 field strain NC20-1. Additionally, differences in sex responses were visible: At 28 dpv, males showed higher PRRSV-2 viremia compared with female pigs (Figure 3C).

Viral shedding in nasal swabs peaked at 28 dpv and then decreased at 49 dpv (Ct < 37) (Appendix A). Out of 60, 9 pigs presented PRRSV-2 RNA in nasal swabs at 28 dpv and only 3 were positive at 49 dpv (Appendix A).

Viral loads in BAL at necropsy were calculated by NanoString. Based on the selected 238 cutoff after normalization, all animals but 2 were positive for PRRSV-M and 9 were positive for PRRSV VR2332 (Appendix A). These data suggest that the lungs of most pigs at 49 dpv were infected with the PRRSV field strain NC20-1 and not the MLV strain.

Overall, based on both NanoString BAL reads and viremia RT-qPCR data, PRRSV-2 NC20-1 was the most prevalent strain detected in PRDC pigs at the end of the experimental layout.

### 3.3. Controlling PRRSV-2 Field Strain Infection Improves Pig Weight Gains During PRDC

While some animals were not able to reduce serum PRRSV-2 field strain loads from 28 to 49 dpv, others were able to control serum PRRSV-2 levels (indicated by a decreasing grey line with a ∆Ct from 28 to 49 dpv of <−2, Figure 3A). We first separated these two fractions (with and without PRRSV-2 field strain “control”) and then analyzed if they show differences in pig health parameters. None of the included parameters were significantly different in these comparisons. While this lack in significant differences included pig weight gains from 28 to 49 dpv (Figure 4A), Spearman correlation analyses revealed that PRRSV-2 field strain control significantly correlated with weight gains between 28 and 49 dpv (Figure 4B). Hence, animals that controlled PRRSV-2 field strain from 28 to 49 dpv had at the same time also a higher weight gain.

### 3.4. Antibody Response to PRRSV Vaccination and Exposure Before and During PRDC

All pigs were PRRSV antibody-negative at the time of placement into the barn and before vaccination with PRRSV-2 MLV. As shown in Figure 5, at 0 dpv, all pigs were negative for serum anti-PRRSV IgG antibodies (S/P < 0.4). By 28 dpv, all pigs developed a systemic antibody response (S/P: average: 1.6, range: 0.9–2.4) which remained consistent at 49 dpv with a slightly higher average (S/P= 1.8) (Figure 5A). At day 49, serum IgG levels also largely correlated with IgG (ρ = 0.751) and IgA (ρ = 0.579) levels in BAL (Appendix A). Hence, serum IgG analysis can provide an accurate estimate of anti-PRRSV IgA and IgG levels at mucosal sites.

Neutralizing antibodies (NAs) against the PRRSV-2 NC20-1 field strain could not be detected at 28 dpv (d.n.s.). After 7 weeks of vaccination, serum NA titers against the field strain ranged from 0 to 1:16 with an average titer of 6; in BAL, they ranged from 0 to 1:64 with an average titer of 7.6 (Figure 5B). Interestingly, intersecting grey lines in Figure 5B demonstrate that serum NA levels were not able to predict NA levels in BAL. These antibody data demonstrate that while vaccination promoted a uniform anti-PRRSV antibody response, NAs appeared late and with two exceptions in BAL, did not reach titer levels > 16 against the PRRSV-2 field strain NC20-1.

### 3.5. Cellular Interferon-γ Response to PRRSV

In contrast to the rather uniform serum IgG levels, the cellular IFN-γ response differed greatly between animals (Figure 5C). While the IFN-γ response in PBMC to PRRSV-2 NC20-1 field strain restimulation was negative at 0 dpv, pigs showed a diverse IFN-γ production at 28 dpv ranging from 36 to 1159 spots per 500,000 PBMC (average: 391). After this peak and by 49 dpv, both the range and average dropped to 12–552 and 182, respectively. These data demonstrate that despite all animals coming from the same farm with same genetic background, and despite all receiving the same vaccine at the same time, the anti-PRRSV cellular immune response varies greatly from animal to animal.

### 3.6. Relations Between Anti-PRRSV Immune Parameters and PRRSV Levels

Quantification of multiple pathogens including MLV-like PRRSV-2 and overall PRRSV-2 was performed via NanoString (Figure 6, bottom two rows: PRRSV-VR-2332 type and PRRSV universal). While most animals cleared MLV-type PRRSV from their lungs by day 51, all but two pigs showed strong, ongoing universal PRRSV-2 lung infections (Appendix A). With the absence of MLV-type PRRSV in most animals, this PRRSV-2 will mostly represent the field strain virus. Relationship analyses between anti-PRRSV immune parameters and PRRSV levels via multivariate analysis were unable to reveal relationships between a systemic IFN-γ response and PRRSV-2 levels or between the anti-PRRSV humoral response with universal lung PRRSV-2 loads. However, the analysis revealed that while BAL IgA (*p* = 0.002) and serum IgG levels (*p* = 0.006) showed a positive correlation with BAL PRRSV MLV-type levels, higher serum NA levels correlated with lower BAL PRRSV MLV-type loads (*p* = 0.024). While the biological background and relevance of the IgA and IgG correlation is unknown, the NA relation shows that animals that induce a stronger systemic NA response clear PRRSV MLV faster.

### 3.7. PRDC-Related Pathogens and Their Relationships with PRRSV

#### 3.7.1. Overall Prevalence of 21 Respiratory Pathogens Involved in PRDC

The prevalence of 21 respiratory pathogens involved in PRDC was quantified in BAL using NanoString technology (Figure 6). In addition, a Sus scrofa gene (ABCF1) was included as control. The control gene showed high enrichment as well as consistent isolation and detection capacity of the NanoString system. Of the 21 pathogens analyzed by NanoString, 15 pathogens stayed on average below the set threshold (238) and six were above threshold—PRRSV, PCMV, PPIV, *B. bronchiseptica*, *G. parasuis*, and *M. hyorhinis*. Of these pathogens, PRRSV-2 had by far the highest BAL gene reads.

For five of these six pathogens, RT-qPCR analyses were available at ISU VDL. To validate NanoString technology as a method for BAL pig pathogen quantification, BAL samples from seven pigs were additionally sent to ISU VDL for qPCR quantification of PRRSV-2, *G. parasuis*, *M. hyorhinis*, PCMV, and PPIV (Table 1). NanoString and ISU VDL data showed positive correlations for all five pathogens. For PRRSV-2, *G. parasuis*, and PPIV, slopes of the linear regression were significant with R^2^ values of 0.88, 0.88, and 0.7, respectively. For *M. hyorhinis*, R^2^ was 0.55 and a *p* value of 0.057. PCMV correlated least with an R^2^ of 0.45 and a *p* = 0.101 (Appendix A). In conclusion, compared to RT-qPCR quantification, NanoString pathogen quantification resulted in a similar trend for PCMV, and provided similar quantification results for *M. hyorhinis* (R^2^ > 0.5) and very similar quantification (R^2^ ≥ 0.7) for PRRSV-2, *G. parasuis*, and PPIV (Appendix A, Table 1).

All six prevalent pathogens were included in the following analyses to determine their relationships with PRRSV-2 prevalence and immunity.

**Figure 6 vaccines-13-00740-f006:**
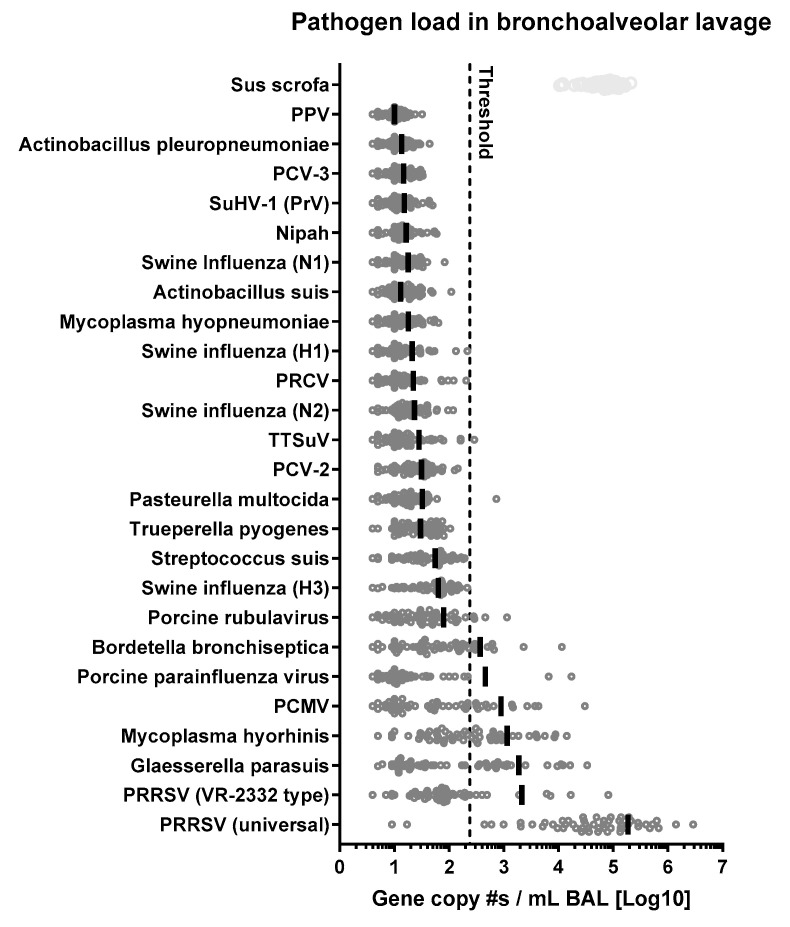
PRDC pathogens detected in bronchoalveolar lavage (BAL) by NanoString. Sus Scrofa gene ABCF1 is used as a control. After data normalization, the detection threshold was set at 238. RNA was extracted from each BAL sample (400 µL), hybridized with the CodeSet for 18 h and analyzed in the NanoString nCounter platform. PPV = porcine parvovirus. PCV = porcine circovirus. SuHV1 or PrV = Pseudorabies virus. PRCV = porcine respiratory coronavirus. TTSuV = porcine torque teno virus. PCMV = porcine cytomegalovirus. PRRSV VR2332 = PRRSV-2 MLV strain. PRRSV M universal for all strains.

#### 3.7.2. *Glaesserella* (*Haemophylus*) *parasuis*

Bacteria loads in BAL were negatively related with the systemic anti-PRRSV IFN-γ response at 49 dpv (*p* = 0.056) but also with PRRSV-2 MLV-like viremia at day 28 (*p* = 0.059): a higher bacterial load correlated with low PRRSV-2 MLV-like viremia. While the direct conclusion requires discussion, this result indicates that PRRSV and/or PRRSV immunity can modulate *Glaesserella parasuis* infection.

#### 3.7.3. *Mycoplasma hyorhinis*

This bacterium showed positive correlation with anti-PRRSV IgA (*p* = 0.003), PRRSV-2 (universal, *p* = 0.04) and PRRSV-2 MLV-like (*p* = 0.007) levels in BAL. This result indicates that PRRSV-2 promotes *Mycoplasma hyorhinis* infection.

#### 3.7.4. Porcine Cytomegalovirus

This virus negatively related with BAL anti-PRRSV IgG (*p* = 0.019) and positively correlated with PRRSV-2 viremia (coinfection with MLV and field strain) at day 28 (*p* = 0.017) and serum anti-PRRSV-2 IgG levels at 49 dpv (*p* = 0.019). Additionally, PCVM had a negative relationship (*p* < 0.001) with weigh gain at day 49: higher levels of PCMV correlated with lower weight gain. While the reason behind the correlation with antibodies is unclear, this result once more indicates that PRRSV-2 can promote the propagation of another pig pathogen—this time PCMV—and PCMV has a detrimental effect on the overall health.

#### 3.7.5. Porcine Parainfluenza Virus (PPIV)

As most previous pathogens, PPIV positively correlated with PRRSV-2 reads, this time with PRRSV-2 MLV-like BAL levels (*p* < 0.001). This result indicates that the PRRSV-2 MLV vaccine strain infection positively affects PPIV propagation.

#### 3.7.6. *Bordetella bronchiseptica*

This bacterium positively correlated with BAL PRRSV-2 loads—both universal and MLV-like (*p* = 0.001 and *p* < 0.001, respectively). The data confirm the synergism between PRRSV-2 and *Bordetella bronchiseptica* infections.

#### 3.7.7. Interaction of *M. hyorhinis* and *B. bronchiseptica*

Apart from the relations between PRRSV-2 and lung pathogens involved in PRDC and PRRS, there was only one correlation between two other pathogens: *M. hyorhinis* and *B. bronchiseptica* positively related with a correlation coefficient of ρ = 0.620 (Appendix A). This indicates that *M. hyorhinis* and *B. bronchiseptica* exert synergistic effects.

### 3.8. Influence of Pathogen Loads and PRRSV Immunity on PRDC Severity

Within the six prevalent pathogens, two pathogens significantly correlated with negative clinical outcomes: PCMV and PRRSV-2.

PCMV has a highly significant negative relation with weight gains from day 28 to 49 (*p* < 0.001). This indicates that PCMV infection affects negatively pig health.

PRRSV-2 showed several relations with PRDC: (i) PRRSV-2 MLV-like viremia levels at day 28 positively related with sick scores (*p* = 0.002); (ii) PRRSV-2 MLV-like presence in BAL at day 51 correlates with a lower temperature at day 49 (*p* = 0.039); (iii) high PRRSV-2 viremia levels at day 28 correlate with lower temperatures at day 49 (*p* = 0.031); (iv) PRRSV-2 viremia at day 49 negatively relates with weight gains (*p* = 0.055); (v) PRRSV-2 viremia levels at day 49 correlate with higher temperatures at that time point (*p* = 0.001). This shows that after PRRSV-2 MLV vaccination, an early PRRSV-2 MLV-like propagation (by day 28) and later its presence in BAL, correlate with decreased PRDC sick scores and fever. Furthermore, early PRRSV-2 viremia (at day 28) does positively affect downstream temperatures at day 49. In contrast, PRRSV-2 viremia at day 49, which is almost exclusively field strain PRRSV-2, impairs weight gain (from 28 to 49 dpv) and leads to higher body temperatures at the same time point. This demonstrates that PRRSV-2 vaccination can positively influence PRDC outcomes and that PRRSV-2 field strain infection has a significant negative impact on pig weight gains and health.

## 4. Discussion

“Nothing’s ever easy.” This quote from the wizard Zeddicus Zu’l Zorander does not only apply in the fantasy novel “Sword of Truth” [35] but also to researchers working in complex diseases like PRDC.

We designed this field trial experiment with two clear hypotheses in mind: (i) based on the central role of PRRSV in PRDC, we expected the vaccine-induced anti-PRRSV response to correlate with clinical PRDC outcome; and (ii) within the 21 analyzed lung pathogens, some pathogens or pathogen groups, explain/correlate with lung disease outcome. Thereby, we wanted to decipher not only the effect of a favorable vaccine-induced anti-PRRSV immune response but also of secondary agents on PRDC outcome. Nevertheless, despite extensive planning, diligent sample collection, and a wealth of data on health, immunity, and pathogen prevalence, the complex interplay between pathogens and immune responses in PRDC remains to be fully unraveled: while we offer valuable insights for future research this also highlights the need for continued investigation into this relevant disease complex.

This study provides at least ten valuable lessons: (i) the uncontrolled settings of field trials complicate data interpretation; (ii) a “healthy” (high) weight at weaning promotes future growth and disease resilience; (iii) most pigs clear PRRSV MLV strain within 7 weeks; (iv) in our study, PRRSV-2 field strain was the most prevalent lung pathogen during PRDC; (v) controlling this PRRSV-2 field strain leads to higher weight gain; (vi) without a well-defined, lung-focused scoring system, overall clinical scoring was not able to predict gross and histological lung pathology; (vii) despite the induction of anti-PRRSV antibodies, the MLV did not lead to field strain-specific NAs at 4 weeks post vaccination and even at 7 weeks post vaccination, NA levels against the PRRSV-2 field strain were low; vii) while serum anti-PRRSV-2 levels were consistent but most likely largely not targeted against the field strain, the cellular immune response was variable but included strong systemic IFN-γ production against the PRRSV-2 field strain; (viii) NanoString quantification of pig pathogens is a solid method to monitor pig pathogens; (ix) most detected lung pathogens correlated with PRRSV-2 viremia or lung loads; and (x) out of the 21 analyzed and 6 detected pig pathogens, two viruses significantly correlated with the severity of the clinical outcome of PRDC–PRRSV-2 and PCMV. In the following, some of these lessons will be put into context with the literature.

The observed positive effect of pig weaning weight on pig growth and disease resilience in the finisher phase has previously been reported: Collins et al. showed that weaning weight influences average daily gain and average daily feed intake until the end of the weaner phase and weaner mortality is greatest in light pigs compared with those weaned at the heaviest weight [36]. Heavy weaning weight piglets have improved post-weaning growth performance and carcass lean measurements compared with light weaning weight piglets [37]. Weight at weaning has a profound influence on carcass weight and it is a major determinant of lifetime growth performance [36].

Surprisingly, overall clinical scoring was not able to predict histological lung pathology in our study. This lack in correlations is not consistent with previous studies performed by Wagner et al. where 8–11-week-old pigs were infected with PRRSV-2 VR2332 [38]: these pigs showed fever, coughing, reduced appetite and respiratory distress, with peripheral airway obstruction, increased respiratory rate and decreased tidal volume that correlated with mild to moderate multifocal areas of interstitial pneumonia. Similarly, Pessoa et al. [39] found positive associations between prevalence of pneumonia and coughing and respiratory distress index in finishers. They used sound recording (SOMO box, SoundTalks^®^, Leuven, Belgium) to monitor coughing in the animals for 13 weeks. Besides the often-argued difference in PRRSV-2 strains, the most likely explanation for this discrepancy is our lack of a detailed assessment of pulmonary health. To effectively handle the assessment and following euthanasia of the 60 pigs, general health assessment was scored on a less lung-focused and more general health scoring system. This certainly represents a limitation of this study. Additionally, age differences and the pathogens milieu might have influenced clinical signs in our field layout compared to a single PRRSV experimental infection and evaluation of respiratory disease in finishers. Other potential explanations for the differences might be the one-time health assessment compared to a weekly/daily monitoring in experimental layouts. In addition, in nearly all animals and despite the absence of fever, most lungs showed gross pathological lesions during the presumed PRDC period. In conclusion, future studies should include a well-defined and target-organ-oriented scoring system.

In addition to health assessments and pig weights, the anti-PRRSV-2 antibody response was quantified longitudinally in serum and in BAL at necropsy (51 dpv). While vaccination promoted a uniform anti-PRRSV antibody response, neutralizing antibodies (NAs) appeared late and, except for two pigs, BAL NA levels against the PRRSV-2 NC20-1 field strain were low-NA titers ≤16. Serum NA levels inversely correlate with PRRSV-2 MLV levels in the lung; in addition, animals with higher NA responses were able to clear the MLV strain faster. This late occurrence of MLV-induced NAs and their limited cross-protection against field strains supports previous findings from Kick et al.: while both field and MLV strains could induce heterologous cellular immune responses peaking mostly at four weeks post inoculation [24], the humoral immune response upon MLV-strain inoculation was delayed and mostly homologous: while both field strains induced homologous and heterologous NAs within 1–3 weeks, the MLV PRRSV-2 strain induced only low-level homologous NAs after 7 weeks; more importantly, over the whole nine-week trial, the MLV failed to induce relevant serum levels of heterologous NAs against the NC134 and NC174 field strains [40]. The current field trial as well as the previous experimental trial stress two important points for anti-PRRSV immunity: (i) vaccine-induced heterologous NAs appear long after vaccine strain clearance; (ii) vaccine-induced NA mostly lack neutralizing capacity against heterologous field strains. This knowledge provides an additional lesson for PRRSV vaccinology: to protect growing pigs, vaccines should target the induction of a more heterologous cellular immune response. This is also supported by Proctor et al. who identified the systemic anti-PRRSV CD4 T-cell response as a promising immune correlate of protection [41]. For maternal-derived immunity, however, NAs seem to play an important role: in a combined field/experimental animal trial, Kick et al. showed that for PRRSV-2, NAs are the driving factor of maternal-derived immunity [42]. In this Kick et al. trial, in addition to standard MLV vaccination, sows received one of two autogenous inactivated virus (AIV) vaccines. Piglets derived from the AIV-vaccinated sows had lower viral loads and significantly lower lung pathology. Between group comparisons of the sow and piglet cellular and humoral immune response showed two things: (i) the cellular immune response in sows and piglets were similar between all groups; (ii) while piglets from the MLV-only vaccinated sows did not show any NAs against the field strain, piglets from the AIV-vaccinated sows were not only better protected from the field strain but also had high serum NA levels against the field strain. This shows that while NAs seem secondary for protecting growing pigs, NAs seem to be essential for maternal-derived immunity. Nevertheless, maternal-derived NAs act against homologous strains emphasizing the need for AIV vaccination of sows to protect their piglets against PRRSV-2 field strains [42].

Besides the assessment of pig health and the influence of PRRSV-2 immunity, this study also determined the effects of 21 pig lung pathogens on PRDC. To this end, we introduced NanoString technology for the quantification of lung pig pathogens. Although only 12.7% of the total BAL samples were validated by ISU VDL, they represented a diverse range of expressions for the target pathogens. This novel use of NanoString technology showed very promising results: For all five pathogens for which molecular tests were available, NanoString results positively correlated with qPCR results with R^2^ values over 0.5 for four out of five pathogens. No molecular test was available for *B. bronchispetica* at ISU VDL. So far, previous studies established multiplexed qPCR approaches for up to six pathogens: PRRSV, CSFV, and PCV2 in [43], CSFV, ASFV, PCV2, PRRSV, and PPV in [44], and PCV2, PRRSV, PRV, CSFV, PPV1 and Japanese encephalitis virus (JEV) in [45]. Recently, Zhou et al. used a multiplex ligation-dependent probe amplification assay for pig pathogen quantification [46], and Lung et al. developed an electronic microarray assay including 30 probes for viruses and 25 probes for bacteria to evaluate PCV2, PRCV, PRRSV, SwIAV, *M. hyopneumoniae*, *P. multocida*, *S. suis*, *Salmonella cholerasuis*. The microarray approach was compared with a multiplex RT-PCR and the analytical sensitivity of the electronic microarray assay was reduced only for some of the targets [47]. One of most recent multiplex assays was developed by Goto et al. where primers were designed for 9 viruses (CSFV, SwIAV, PRCV, suid herpesvirus 1, PRRSV-2, PRRSV-1, PCV2, PCV3, PCMV) and 7 bacteria (*M. hyopneumoniae*, *A. pleuropneumoniae*, *P. Multocida*, *G. parasuis*, *Salmonella* spp., *S. suis*, *B. bronchiseptica*) [48]. At the same time, next generation sequencing (NGS) [49] and Oxford Nanopore technologies have been evaluated in profiling the presence of different pathogens in swine herds [50,51,52]. While these established multiplex assays and NGS technology can be useful, the potential to detect up to ~400 pathogens and the ease of the involved handling and data analysis make the NanoString technology very valuable for studying pig pathogens in complex multifactorial diseases like PRDC.

Our findings reveal that PCMV was positively correlated with PRRSV-2 viremia in coinfected pig at 28 dpv, consistent with prior research demonstrating a positive association between PRRSV and PCMV (*p* < 0.05) [53]. Critically, PCVM presence negatively correlated with weight gain at day 49, suggesting a detrimental impact on pig clinical outcome.

*G. parasuis* in BAL showed non-significant negative correlations with PRRSV-2 MLV-type viremia at 28 dpv (*p* = 0.059) and with an anti-PRRSV IFN-γ response at 49 dpv (*p* = 0.056); however, no negative correlations with field-strain PRRSV-2 were observed. Given the fact that MLV-type infected pigs in serum at 28 dpv and in BAL at 51 dpv, could limit PRDC illness scores and fever, a possible explanation is that a successful PRRSV-2 MLV vaccination improved the overall health of pigs; in turn, this could have limited the growth of *G. parasuis*. Hence, while a direct positive correlation between *G. parasuis* and PRRSV-2 field strain is lacking in our study, it is not in contrast with previous studies that showed a synergic effect of the two pathogens in regard to pathogen propagation and/or animal health—as described in [21,54,55,56,57], and reviewed in Saade et al. [6]. More mixed results were shown by Segales et al.: While disease severity increased in dually infected pigs compared with virus alone, there was no influence of previous PRRSV-2 VR2332 infection on the occurrence of *G. parasuis* infection [58]. Authors explained inconsistencies with other studies by proposing a difference in the virulence of PRRSV isolates.

*M. hyorhinis* infection was promoted by the prevalence of PRRSV-2 in BAL. This outcome is supported by different publications [54,59,60] and reviewed in Saade et al. [6]. Lee at al. showed a positive correlation between PRRSV and *M. hyorhinis* with exacerbated lung lesions in coinfected animals. Additionally, percentages of seroprevalence in seven pig herds showed that PRRSV and *M. hyorhinis* was the most prevalent coinfection in the field [60]. The coinfections caused severe pneumonia [58], increased the severity of polyserositis, and the odds of detecting *M. hyorhinis* genome fragments were 3.63 times higher in PRRSV positive animals [54]. Hence, so far, the literature suggests synergistic effects of PRRSV-2 and *M. hyorhinis* with detrimental health outcomes.

This study also showed a positive correlation between PPIV-1 and PRRSV-2 in BAL highlighting a potential synergism between the PRRSV vaccine strain and PPIV-1 during PRDC. Very few studies have been evaluating the relationship between PPIV-1 (also called porcine respirovirus 1) and other pathogens. Gauger’s group was the first to evaluate PPIV-1 infection in the context of SwIAV respiratory disease. They performed an experimental infection comparing PPIV-1 and swine IAV single infections with a dual challenge that contained an equal volume of PPIV-1 and swine IAV and virus titers equivalent to single infection. Coinfection did not appear to exacerbate clinical signs compared to their respective PPIV-1 or SwIAV individually challenged groups. Coinfected and SwIAV groups showed significantly higher macroscopic pneumonia and microscopic lung lesion scores compared to the PPIV-1 and control pigs. Replication in the lower respiratory tract and upper one in coinfected pigs was either similar to, or lower than the pigs challenged with only SwIAV or PPIV-1 [61]. A study in Poland showed that PPIV-1 was highly prevalent in the 30 selected herds and, although coinfection with IAV (23.5%) and PRRSV-1 (11.8%) were infrequent, they correlated with the presence of respiratory clinical signs [62]. A study in Italy described frequent PRRSV and PPIV-1 coinfections in pigs with respiratory symptoms [63]. In summary, the few studies (including the present one) support a synergistic detrimental role for PRRSV-2/PPIV-1 field coinfections.

*B. bronchiseptica* loads correlated with both PRRSV universal and MLV-like reads in BAL highlighting a synergism between the two pathogens. This is supported by a previous experimental infection where 3-week-old pigs were inoculated intranasally with single infections or coinfection of PRRSV-2 NADC-21 and *B. bronchiseptica* KM22. Clinical signs, fever and decreased weight gain were more severe in the coinfection group that showed lesions consistent with bronchopneumonia [64]. Additionally, our field study showed that *M. hyorhinis* and *B. bronchiseptica* exert a potential synergistic effect. This is consistent with a previous experimental coinfection of gnotobiotic piglets. Pigs developed severe clinical disease, and the death rate increased compared with the individually challenged piglets [65]. The current work supports what Gois et al. hypothesized at the end of their study in gnotobiotic pigs: A similar pathogenic synergism would be possible under natural conditions, even though concurrent infections in the field could show higher variability related to the pathogens milieu and the presence of maternal antibodies.

The ten key lessons derived from this study, ranging from the challenges of field trial interpretation to the potential synergistic effect of specific pathogen combinations, collectively provide a valuable framework for future research aimed to further unravel the complex interplay of factors driving PRDC. Ultimately, while a simple and conclusive answer to the multifaceted nature of PRDC remains elusive, this study offers a significant step forward by highlighting key areas of focus and the need for refined diagnostic tools for effective intervention strategies.

## 5. Conclusions

While our meticulously designed field trial yielded a wealth of data and provided at least ten valuable lessons regarding PRDC pathogenesis, the intricate dance between the porcine immune system and the diverse array of respiratory pathogens remains a complex puzzle. We have highlighted the beneficial impact of high weaning weight, the dynamics of PRRSV clearance and antibody responses, and the limitations of clinical scoring. Furthermore, this study underscores the potential of NanoString technology for comprehensive pathogen surveillance and identified PRRSV-2 and PCMV as key viral players in the clinical outcome. The observed correlations between PRRSV-2 and other pathogens like PCMV, *M. hyorhinis*, PPIV-1, and *B. bronchiseptica* further emphasize the multifactorial nature of this disease complex and the potential for synergistic interactions. Ultimately, this research contributes valuable pieces to the PRDC puzzle, but the pursuit of effective PRDC control undoubtedly requires continued innovative research.

## Figures and Tables

**Figure 1 vaccines-13-00740-f001:**
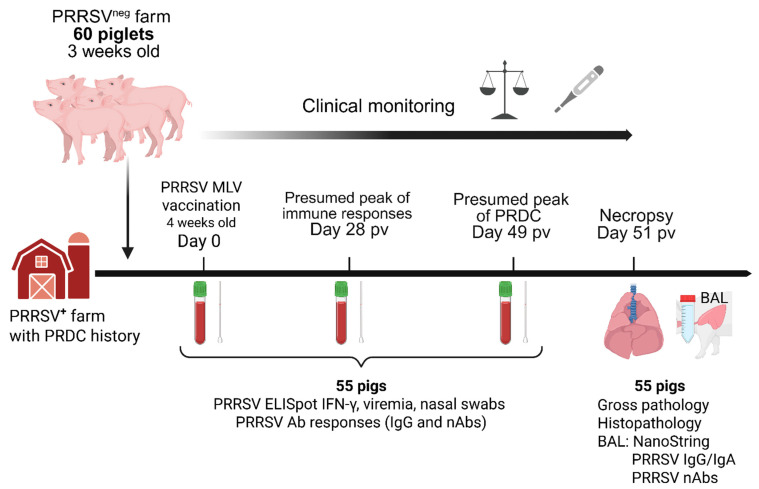
Study design. Sixty commercial Yorkshire 3-week-old piglets balanced by sex were selected from a PRRSV-negative herd and introduced into a farm with history of porcine respiratory disease complex (PRDC). Animals were vaccinated intramuscularly with 2 mL of Ingelvac^®^ PRRSV-2 modified live virus (MLV) vaccine at 4 weeks of age. Serum, whole-blood, and nasal swabs were collected prior to vaccination (0 days post vaccination, dpv), at 28 dpv and when PRDC was expected (49 dpv). At 51 dpv, animals were euthanized, and lungs were harvested. Lungs were flushed with 1× PBS to collect the bronchoalveolar lavage (BAL), and a total of five lung samples were taken for histopathology. ELISpot IFN-γ, viremia, shedding and antibody responses were analyzed at day 0, 28 and 49. NanoString and antibody responses were performed in BAL at 51 dpv. Partial sampling was performed on five pigs that succumbed during the experimental process. Created in BioRender. Crisci, E. (2025) https://BioRender.com/1r8uij0.

**Figure 2 vaccines-13-00740-f002:**
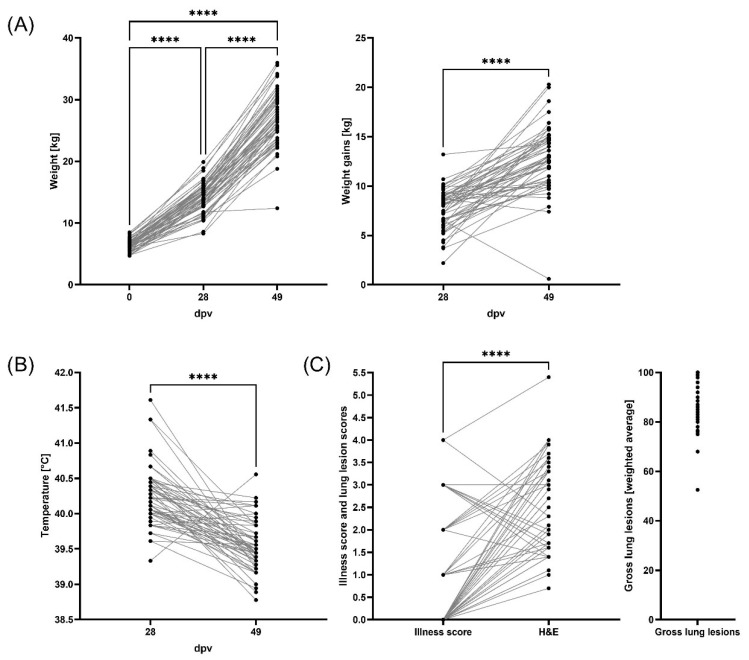
Clinical monitoring and lung lesions. Pigs were monitored for illness, temperature and weight before MLV PRRSV vaccination and at 28 and 49 days post vaccination (dpv). At the end of the trial, pigs were evaluated for size, coughing, fitness and illness (clinical score) and at 51 dpv pig were euthanized and lung collected for macroscopic lung lesions and histopathology evaluation. (**A**) Weight (kg) and average weight gain (kg) at 0, 28 and 49 dpv. (**B**) Body temperatures (°C) were evaluated at 0, 28 and 49 dpv. (**C**) An illness score of 0 refers to healthy and 4 to severe clinical disease. Lung histopathology score (H&E) is based on the severity of the interstitial pneumonia, where 0 refers to no microscopic lesions and 6 to severe, diffuse interstitial pneumonia. Gross lesion scores evaluated the percentage of lungs affected with pneumonia. **** *p* < 0.0001.

**Figure 3 vaccines-13-00740-f003:**
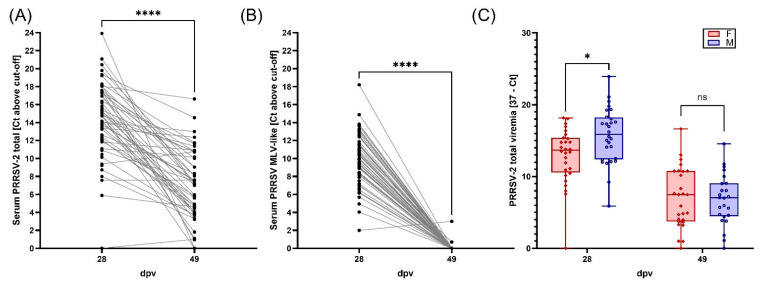
PRRSV-2 MLV and field strains viremia. PRRSV-2 viral load was evaluated in serum at days 28 and 49 dpv by RT-qPCR. (**A**) General PRRSV-2 viremia including both MLV and field strains. (**B**) MLV strain viremia at 28 and 49 dpv. (**C**) General PRRSV-2 viremia in female and male pigs at 28 and 49 dpv. Data are expressed in the format of 37 minus sample Ct (37-sample Ct). * *p* < 0.05, **** *p* < 0.0001, and ns: not significant.

**Figure 4 vaccines-13-00740-f004:**
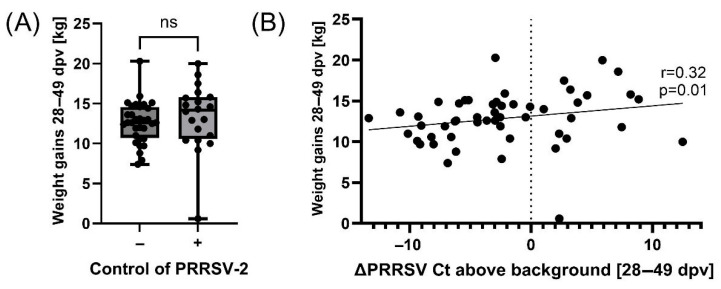
Correlation analyses of PRRSV-2 viremia with weight gain. (**A**) Weight gains of pigs that did not control or did control PRRSV-2 infection (ΔCt from 28 to 49 dpv < 2). (**B**) Spearman correlation analysis between weight gains and the control of PRRSV-2 infection as defined by a reduction in Ct levels above background from 28 to 49 dpv. Statistical analysis was performed using an unpaired *t*-test. ns = *p* > 0.05.

**Figure 5 vaccines-13-00740-f005:**
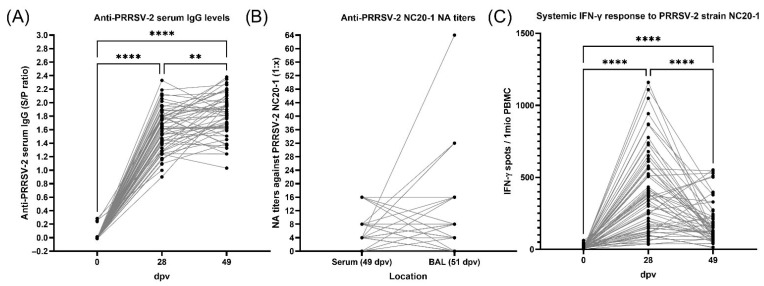
Anti-PRRSV-2 immunity in the context of PRDC. (**A**) Systemic IgG antibody responses against PRRSV-2 at 0, 28 and 49 dpv. Serum PRRSV-2 antibody levels are expressed in S/P ratio values. (**B**) Neutralizing antibodies titers against field strain NC20-1 in serum (49 dvp) and bronchoalveolar lavage (BAL, 51 dpv). Titers are shown as reciprocal of the dilution factor. (**C**) Interferon-γ responses to PRRSV-2 field strain NC20-1 at 0, 28 and 49 dpv. Statistical analysis was performed using a mixed-effects analysis with Tukey’s multiple comparisons test. ** *p* < 0.01 and **** *p* < 0.0001.

**Table 1 vaccines-13-00740-t001:** NanoString and ISU VDL data. qPCR Ct ≥ 35, 37, and 38 are negative. NanoString normalized reads <238 are negative.

Pig	PRRSV-2 M (General)		Glaes. Parasuis		Myco. Hyorhinis		Porcine Cytomegalovirus (PCMV)		Parainfluenza Virus (PPIV)		Swine Influenza (General)		PRRSV-1
	NanoString Normalized Reads	ISU Ct	NanoString Normalized Reads	ISU Ct	NanoString Normalized Reads	ISU Ct	NanoString Normalized Reads	ISU Ct	NanoString Normalized Reads	ISU Ct	NanoString Normalized Reads	ISU Ct	ISU Ct
12	78,491	30.3	494	29.9	733	27.8	64	27.2	16	≥38	111	≥38	≥37
18	44,172	30.1	814	30.1	838	31.3	473	30.9	17,204	24.1	74	≥38	≥37
22	20,863	28.6	9550	26.7	8795	30.4	18	≥35	6601	24.5	108	≥38	≥37
28	523,774	27.6	48	≥35	4038	30.7	24	≥35	24	≥38	48	≥38	≥37
36	162,850	26.6	70	≥35	60	34.7	4320	26.8	40	33.1	120	≥38	≥37
47 Neg	17	≥37	9	≥35	9	32.9	4	≥35	4	≥38	4	≥38	≥37
48	9615	30	5	≥35	5	≥35	9	30.0	5	≥38	5	≥38	≥37

**Table 2 vaccines-13-00740-t002:** Viremia. Viral loads at days 28 and 49 for PRRSV-2 PCR screening. Viral loads at day 28 for PRRSV-2 Ingelvac MLV vaccine-like PCR. qPCR Ct ≥ 37 are negative. Ct difference of ≥5 between the PRRSV-2 screening PCR and the Ingelvac MLV vaccine-like PCR was used to infer coinfection of both the MLV vaccine virus and field strain NC20-1.

Pig	PRRSV-2 Ct D28	PRRSV-2 Ct D49	MLV-Like Ct D28	Ct MLV − Ct PRRSV-2 D28	Pig	PRRSV-2 Ct D28	PRRSV-2 Ct D49	MLV-Like Ct D28	Ct MLV − Ct PRRSV-2 D28
1	22.20	26.20	23.84	1.63	29	27.63	26.23	29.09	1.46
2	24.93	32.59	26.39	1.46	31	17.77	28.95	26.15	8.38
3	20.79	33.21	30.47	9.68	32	21.16	33.78	27.09	5.92
4	29.00	33.69	30.85	1.86	33	20.04	31.05	24.22	4.19
5	24.86	N/A	26.06	1.20	34	19.63	26.23	24.45	4.82
6	26.09	29.51	27.38	1.29	35	22.90	25.62	27.69	4.79
7	21.72	32.17	23.41	1.68	36	18.95	20.37	27.35	8.40
8	18.97	28.95	26.57	7.60	37	23.48	32.08	25.39	1.91
9	21.95	31.30	23.19	1.24	38	13.08	≥37	25.56	12.49
10	24.38	28.67	25.84	1.46	39	25.91	26.83	28.01	2.10
11	25.15	26.08	26.67	1.52	40	22.85	33.12	30.03	7.17
12	23.23	29.52	25.55	2.32	41	23.20	29.42	24.59	1.39
13	≥37	29.35	≥37	0.00	42	21.75	29.95	28.70	6.95
14	22.27	35.19	27.92	5.65	43	19.69	27.95	26.30	6.61
15	26.55	N/A	27.74	1.19	44	23.73	32.31	25.44	1.71
16	24.50	N/A	25.86	1.36	45	19.63	29.55	30.00	10.36
17	17.66	22.45	18.80	1.14	46	26.85	33.35	28.07	1.23
18	22.94	24.00	25.13	2.19	47	22.21	≥37	24.51	2.30
19	18.85	36.04	26.98	8.13	48	19.42	26.98	23.99	4.57
20	24.65	31.40	27.71	3.06	49	31.11	32.39	32.06	0.95
21	18.72	25.26	27.59	8.88	50	23.59	24.64	28.83	5.24
22	28.27	33.27	29.82	1.54	51	27.78	N/A	29.49	1.71
23	20.06	33.03	28.65	8.59	52	22.23	32.94	23.73	1.49
24	16.56	29.99	25.59	9.03	53	15.92	29.81	31.31	15.39
25	19.54	35.90	28.10	8.56	54	20.63	25.34	22.12	1.49
26	25.09	31.10	26.56	1.47	55	25.00	32.28	32.96	7.96
27	21.44	N/A	30.31	8.87	56	21.60	26.29	23.24	1.64
28	29.43	29.39	30.76	1.33	59	17.20	27.97	28.55	11.35

## Data Availability

The original contributions presented in this study are included in the article/Appendix A. Further inquiries can be directed to the corresponding author.

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
