# Peer review of "Challenges and Lessons Learned from a Field Trial on the Understanding of the Porcine Respiratory Disease Complex"

_vaccines, 2025, doi:10.3390/vaccines13070740_

Round 1
Reviewer 1 Report
Comments and Suggestions for Authors
The manuscript entitled “Challenges and Lessons Learned from a Field Trial on the Understanding of the Porcine Respiratory Disease Complex” investigated the factors influencing Porcine Respiratory Disease Complex (PRDC) severity in pigs vaccinated against PRRSV. PRRSV-negative weaners received a modified live virus (MLV) vaccine and entered a PRDC-affected farm. Researchers monitored anti-PRRSV immune responses before and after vaccination and during a subsequent PRDC outbreak. Analysis of lung fluid (BAL) at necropsy detected PRRSV in most pigs alongside five co-pathogens (notably Porcine cytomegalovirus/PCMV). Key findings showed that higher weaning weight predicted better disease resilience and PRRSV-2 field strain control. While the vaccine MLV strain was cleared by most pigs within 7 weeks, the PRRSV-2 field strain became the dominant lung pathogen. Vaccine-induced neutralizing antibodies against this field strain were limited and slow to develop, though a systemic cellular (IFN-γ) response occurred. Critically, the loads of PRRSV-2 and PCMV in the lungs were significant predictors of PRDC severity. The study highlights the complex interplay between vaccination, co-infections (especially PRRSV-2 and PCMV), and host factors in PRDC outcomes. However, several shortcomings in the study need to be addressed to improve clarity and substantiate the conclusions drawn.
Major issues:
- The abstract mentions "PRRS MLV" vaccination and later refers to "PRRSV-2 field strain". What specific genotype (PRRSV-1 or PRRSV-2) was the vaccine strain? How was the "field PRRSV strain" identified and differentiated from the vaccine strain? Was sequencing or strain-specific PCR used? This is critical for interpreting points (ii) and (iv).
- Points (i), (vi), and (vii) state correlations and predictive The term "pathobiont" is used for the co-detected agents. While technically acceptable, it might be less common than "pathogen" or "infectious agent" in this context for some readers. Consider consistency or defining it briefly if used.
- In the introduction, paragraphs 3 to 7 provide a detailed enumeration of pathogen prevalence percentages and associations across multiple countries and regions (the United States, Denmark, France, Italy, Austria, Brazil, China, and India). While providing background is necessary, these details are overly cumbersome and deviate from the specific focus of this study (PRRSV vaccination, immune responses, and secondary infections under field conditions). It is recommended to substantially streamline the global overview section. Focus on summarizing key consensus points relevant to the study’s hypotheses. Avoid listing extensive specific percentages and regional variations unless they directly justify the research rationale or pathogen selection criteria. If absolutely necessary, excessive details may be relocated to supplementary tables.
- The stated objective in Paragraph 10 ("...this study aims to evaluate the immune response to PRRSV vaccination and the role of PRRSV and secondary infections...") is overly broad. Although subsequent sentences outline the methodology, key immunological questions—such as the quality/kinetics of neutralizing antibodies, cell-mediated responses (e.g., IFN-γ), mucosal versus systemic immunity, or correlations between specific immune parameters and pathogen load/disease severity—are not explicitly addressed upfront. It is recommended to refine and clarify the primary objectives to focus on these critical aspects.
- Although the Materials and Methods section mentions the use of MLV-specific PCR to detect the vaccine strain (Section 2.4), it does not clearly specify how immune responses induced by vaccination were distinguished from those caused by field strain infection.
- The validation of NanoString pathogen detection is inadequate. Only 7 BAL samples (1 negative, 6 positive) were used for RT-qPCR confirmation (Section 2.8, Table 1), representing a small subset (n=7 vs. total >50) without clear selection criteria. Moreover, validation covered only 6/21 targeted pathogens (e.g., PRRSV-2), omitting others like PPIV and B. bronchiseptica.
7 Section 3.8 claims that "PRRSV-2 MLV vaccination can limit PRDC sick scores through early vaccine strain proliferation," but the data in Section 3.2 show that at 28 dpv, 55/56 pigs exhibited vaccine strain viremia, with 21 co-infected with field strains (Table 2), failing to demonstrate the protective effect of the vaccine strain alone on the disease.
- Section 3.4 emphasizes "low levels of neutralizing antibodies (NA)" (Figure 5B), but Section 3.6 states that "serum NA levels inversely correlated with BAL PRRSV MLV-type levels," suggesting that higher NA should promote clearance. The term "inversely correlated" here could be misleading (it should actually be "higher NA levels correspond to lower vaccine strain load").
- In discussing the synergistic effects of PCMV and PRRSV (paragraph 13), it is suggested that "low NA levels may facilitate PCMV infection through antibody-dependent enhancement (ADE)" ("low NAs levels... support the potential ADE detrimental effect"). However, this study did not test for ADE (e.g., macrophage infection enhancement assays), nor did it cite direct evidence of ADE in porcine PCMV (reference 54 provides a general description of macrophage susceptibility).
Minor Issues:
- The introduction consistently uses terms such as "pathogens" or "infectious agents." However, the final paragraph abruptly introduces the term "pathobionts" ("various viral and bacterial pathogenic microorganisms") without justification or clarification for this shift in terminology. To maintain conceptual clarity, we recommend adhering to consistent terminology unless a deliberate distinction is required and explicitly explained.
- The abbreviations "FFN assay" (Section 2.6) and "ConA" (Section 2.7) are not defined upon their first use in the text.
- The term "Ficoll-Paque" (Section 2.1) is inconsistently written as "Ficoll Paque" (Figure 1 legend). The hyphen usage is inconsistent. We recommend standardizing the spelling to "Ficoll-Paque" (the manufacturer's standard format) throughout the manuscript.
- In Section 3.7.5, the term "Porcine Parainfluenza Virus (PPIV)" exhibits inconsistent capitalization. We recommend standardizing the formatting—for example, using "Porcine parainfluenza virus (PPIV)" (with only the first letter of the full name capitalized, following standard virological nomenclature).
- "qPCR results with R2s over 0.5" (error plural form), and it is suggested to correct it to "rvalues > 0.5" (statistical symbol specification).
- "AIV" (undefined for the first time). When the abbreviation appears for the first time, the full name should be written: "Autogenetic in activated virus (AIV) vaccines".
Author Response
The manuscript entitled “Challenges and Lessons Learned from a Field Trial on the Understanding of the Porcine Respiratory Disease Complex” investigated the factors influencing Porcine Respiratory Disease Complex (PRDC) severity in pigs vaccinated against PRRSV. PRRSV-negative weaners received a modified live virus (MLV) vaccine and entered a PRDC-affected farm. Researchers monitored anti-PRRSV immune responses before and after vaccination and during a subsequent PRDC outbreak. Analysis of lung fluid (BAL) at necropsy detected PRRSV in most pigs alongside five co-pathogens (notably Porcine cytomegalovirus/PCMV). Key findings showed that higher weaning weight predicted better disease resilience and PRRSV-2 field strain control. While the vaccine MLV strain was cleared by most pigs within 7 weeks, the PRRSV-2 field strain became the dominant lung pathogen. Vaccine-induced neutralizing antibodies against this field strain were limited and slow to develop, though a systemic cellular (IFN-γ) response occurred. Critically, the loads of PRRSV-2 and PCMV in the lungs were significant predictors of PRDC severity. The study highlights the complex interplay between vaccination, co-infections (especially PRRSV-2 and PCMV), and host factors in PRDC outcomes. However, several shortcomings in the study need to be addressed to improve clarity and substantiate the conclusions drawn.
We thank the reviewer for the positive and critical but constructive comments. These comments were very useful to improve the quality of this manuscript. We have included reviewer’s suggestions in the text using the “Track changes” function or highlighting the text in the manuscript. On the following pages, we have responded to each of the reviewer’s comments item by item.
- The abstract mentions "PRRS MLV" vaccination and later refers to "PRRSV-2 field strain". What specific genotype (PRRSV-1 or PRRSV-2) was the vaccine strain? How was the "field PRRSV strain" identified and differentiated from the vaccine strain? Was sequencing or strain-specific PCR used? This is critical for interpreting points (ii) and (iv).
The MLV vaccine strain is a PRRSV-2 strain. We have clarified the PRRSV type in the manuscript. The PRRSV-2 field strain was sequenced for both ORF5 and whole genome sequencing. The genetic information of both MLV and field strain is provided in section “2.3 PRRSV-2 isolation, propagation, titration”.
- Points (i), (vi), and (vii) state correlations and predictive The term "pathobiont" is used for the co-detected agents. While technically acceptable, it might be less common than "pathogen" or "infectious agent" in this context for some readers. Consider consistency or defining it briefly if used.
We have changed the term “pathobiont” to “pathogen” and make it consistent throughout the text.
- In the introduction, paragraphs 3 to 7 provide a detailed enumeration of pathogen prevalence percentages and associations across multiple countries and regions (the United States, Denmark, France, Italy, Austria, Brazil, China, and India). While providing background is necessary, these details are overly cumbersome and deviate from the specific focus of this study (PRRSV vaccination, immune responses, and secondary infections under field conditions). It is recommended to substantially streamline the global overview section. Focus on summarizing key consensus points relevant to the study’s hypotheses. Avoid listing extensive specific percentages and regional variations unless they directly justify the research rationale or pathogen selection criteria. If absolutely necessary, excessive details may be relocated to supplementary tables.
The introduction has been changed accordingly by summarizing the key consensus points and streamlined the global overview section.
- The stated objective in Paragraph 10 ("...this study aims to evaluate the immune response to PRRSV vaccination and the role of PRRSV and secondary infections...") is overly broad. Although subsequent sentences outline the methodology, key immunological questions—such as the quality/kinetics of neutralizing antibodies, cell-mediated responses (e.g., IFN-γ), mucosal versus systemic immunity, or correlations between specific immune parameters and pathogen load/disease severity—are not explicitly addressed upfront. It is recommended to refine and clarify the primary objectives to focus on these critical aspects.
The paragraph in the introduction has been changed accordingly.
- Although the Materials and Methods section mentions the use of MLV-specific PCR to detect the vaccine strain (Section 2.4), it does not clearly specify how immune responses induced by vaccination were distinguished from those caused by field strain infection.
Detection of PRRSV-specific antibodies (IgG and IgA, section 2.5) was performed using an ELISA that detects total antibodies and does not differentiate between responses induced by vaccination and those by field strain infection. Conversely, neutralizing antibodies and IFN-γ responses were specific and assessed against the field strain NC20-1. We have clarified this aspect in sections 2.6 and 2.7.
- The validation of NanoString pathogen detection is inadequate. Only 7 BAL samples (1 negative, 6 positive) were used for RT-qPCR confirmation (Section 2.8, Table 1), representing a small subset (n=7 vs. total >50) without clear selection criteria. Moreover, validation covered only 6/21 targeted pathogens (e.g., PRRSV-2), omitting others like PPIV and B. bronchiseptica.
We agree with the reviewer that the seven BAL samples constitute a small portion (12.7%, 7/55) of the total samples. However, these samples were specifically chosen to represent a diverse range of expressions for the targeted pathogens. Our goal was to validate the NanoString panel against official reference tests available at the Iowa State Veterinary Diagnostic Lab (ISU VDL). While PPIV was successfully validated by ISU VDL (see Table 1, "Parainfluenza virus" column for NanoString and ISU VDL results), a molecular test for B. bronchiseptica was not performed due to the unavailability of a reference test at ISU VDL. We have included this limitation in the discussion.
- Section 3.8 claims that "PRRSV-2 MLV vaccination can limit PRDC sick scores through early vaccine strain proliferation," but the data in Section 3.2 show that at 28 dpv, 55/56 pigs exhibited vaccine strain viremia, with 21 co-infected with field strains (Table 2), failing to demonstrate the protective effect of the vaccine strain alone on the disease.
We thank the reviewer for the suggestion. We have clarified the statement and arranged the 3.8 section.
- Section 3.4 emphasizes "low levels of neutralizing antibodies (NA)" (Figure 5B), but Section 3.6 states that "serum NA levels inversely correlated with BAL PRRSV MLV-type levels," suggesting that higher NA should promote clearance. The term "inversely correlated" here could be misleading (it should actually be "higher NA levels correspond to lower vaccine strain load").
We thank the reviewer for the suggestion. We have clarified the statement and arranged the 3.4 and 3.6 sections.
- In discussing the synergistic effects of PCMV and PRRSV (paragraph 13), it is suggested that "low NA levels may facilitate PCMV infection through antibody-dependent enhancement (ADE)" ("low NAs levels... support the potential ADE detrimental effect"). However, this study did not test for ADE (e.g., macrophage infection enhancement assays), nor did it cite direct evidence of ADE in porcine PCMV (reference 54 provides a general description of macrophage susceptibility).
We thank the reviewer for the comment. We have removed the paragraph from the discussion.
Minor Issues:
- The introduction consistently uses terms such as "pathogens" or "infectious agents." However, the final paragraph abruptly introduces the term "pathobionts" ("various viral and bacterial pathogenic microorganisms") without justification or clarification for this shift in terminology. To maintain conceptual clarity, we recommend adhering to consistent terminology unless a deliberate distinction is required and explicitly explained.
The term “pathobiont” has been replaced by “pathogen” throughout the manuscript.
- The abbreviations "FFN assay" (Section 2.6) and "ConA" (Section 2.7) are not defined upon their first use in the text.
We have added the definition of FFN and ConA in the text.
- The term "Ficoll-Paque" (Section 2.1) is inconsistently written as "Ficoll Paque" (Figure 1 legend). The hyphen usage is inconsistent. We recommend standardizing the spelling to "Ficoll-Paque" (the manufacturer's standard format) throughout the manuscript.
We have changed the text accordingly.
- In Section 3.7.5, the term "Porcine Parainfluenza Virus (PPIV)" exhibits inconsistent capitalization. We recommend standardizing the formatting—for example, using "Porcine parainfluenza virus (PPIV)" (with only the first letter of the full name capitalized, following standard virological nomenclature).
We have changed the text accordingly.
- "qPCR results with R2s over 0.5" (error plural form), and it is suggested to correct it to "rvalues > 0.5" (statistical symbol specification).
We have changed the text accordingly.
- "AIV" (undefined for the first time). When the abbreviation appears for the first time, the full name should be written: "Autogenetic in activated virus (AIV) vaccines".
We have changed the text accordingly.

Reviewer 2 Report
Comments and Suggestions for Authors
The Porcine Respiratory Disease Complex (PRDC) is a multiple respiratory disease. The outbreak of the disease was simulated in a specific pig farm. The research was innovative and quite meaningful.
1.Whether PRRSV-2 field strain (referred as NC20-1) belongs to a highly pathogenic strain or a classic strain type needs to be indicated by the author
- It is suggested that the author adjust the content of Table 1 properly
- Temperature measurement is a dynamic test, preferably for several consecutive days. Could the author explain why it is only measured for two days.
- PRRSV-2 and PCMV significantly predicted the outcome of PRDC???. Are the simulation results consistent with the clinical detection results? It is suggested that the author further strengthen them in the discussion.
Author Response
The Porcine Respiratory Disease Complex (PRDC) is a multiple respiratory disease. The outbreak of the disease was simulated in a specific pig farm. The research was innovative and quite meaningful.
1.Whether PRRSV-2 field strain (referred as NC20-1) belongs to a highly pathogenic strain or a classic strain type needs to be indicated by the author
We thank the reviewer for the comment. At the time of the study (2019-2020), the field strain NC20-1 was classified as 1-4-4 based on RFLP analysis, belonging to lineage L1A. During that period, strains exhibiting the 1-4-4 RFLP pattern were widely regarded as “highly virulent” or “highly pathogenic”. We have therefore incorporated the term "highly virulent" into the text, considering it a more precise descriptor in this context. It is important to note that within the complex etiology of PRDC, various co-infecting pathogens can differentially influence the overall pathogenicity and disease severity in the field.
- It is suggested that the author adjust the content of Table 1 properly
We have changed the table content accordingly.
- Temperature measurement is a dynamic test, preferably for several consecutive days. Could the author explain why it is only measured for two days.
We agree with the reviewer that temperature is a dynamic test, but conducting a 51-days pig trial with 60 animals was inherently labor-intensive and burdensome. Pigs were placed in a commercial farm that was conducting its normal operations. Daily animal husbandry, individual monitoring for health, and multiple sample collection events - each requiring skilled personnel, animal restraint, and immediate processing - demanded significant time and effort from our team, veterinarians and farm personnel. To manage this workload and adhere to stringent biosafety measures, we opted for targeted temperature monitoring during sample collection (rectal temperature at D28, D49) rather than daily measurements. This approach minimizes animal stress from repeated handling, optimized labor allocation, and upheld our biosafety protocols by consolidating handling events. This strategic decision balanced the need for critical data with the practical demands of a large-scale, long-duration pig study.
- PRRSV-2 and PCMV significantly predicted the outcome of PRDC???. Are the simulation results consistent with the clinical detection results? It is suggested that the author further strengthen them in the discussion.
The multivariate statistical analysis evaluated the outcome of other pathogens levels (G. parasuis, M hyorhinis, PCMV, PPIV, B bronchiseptica) using the predictors of PRRSV immunity measures, PRRSV loads (viremia, BAL, MLV vs field strain). Another multivariate analysis evaluated the outcome of PRDC severity (based on sick score, HE, weight gain and temp at D49) using the predictors of PRRSV immunity measures, PRRSV loads (viremia, BAL, MLV vs field strain) and pathogens levels. Those analyses revealed that pigs coinfected with PRRSV field strain and MLV on day 28 have statistically significantly higher levels of PCMV. Additionally, PCMV has a negative and statistically significant relationship with weight gain on day 49, and therefore on general health. We have clarified this aspect and modified the manuscript in the abstract, results and discussion sections.

Reviewer 3 Report
Comments and Suggestions for Authors
The study by Crisci et al. provides a comprehensive analysis of the complex interplay between the porcine immune system and respiratory pathobionts during the Porcine Respiratory Disease Complex (PRDC). The authors have meticulously designed a field trial to investigate the impact of PRRSV vaccination and various secondary pathogens on PRDC severity. The use of NanoString technology for pathogen quantification is innovative and adds value to the study. Overall, the manuscript is well-written and the methodology is rigorous. However, there are a few areas that require clarification and improvement to enhance the manuscript's impact.
Specific Comments
- The scientific value and significance of this study need to be reflected in the abstract.
- The abbreviations in Figure 1 need to be annotated. Additionally, there is an inconsistency between 60 pigs and 55 pigs; please explain.
- Please provide the animal ethics review and approval number, as it is currently lacking.
- Please modify the format of the references to comply with the journal's requirements.
Author Response
The study by Crisci et al. provides a comprehensive analysis of the complex interplay between the porcine immune system and respiratory pathobionts during the Porcine Respiratory Disease Complex (PRDC). The authors have meticulously designed a field trial to investigate the impact of PRRSV vaccination and various secondary pathogens on PRDC severity. The use of NanoString technology for pathogen quantification is innovative and adds value to the study. Overall, the manuscript is well-written and the methodology is rigorous. However, there are a few areas that require clarification and improvement to enhance the manuscript's impact.
Specific Comments
- The scientific value and significance of this study need to be reflected in the abstract.
We have changed the abstract accordingly.
- The abbreviations in Figure 1 need to be annotated. Additionally, there is an inconsistency between 60 pigs and 55 pigs; please explain.
We have changed the text accordingly. Section 2.1 describes the reason of the inconsistency between 60 and 55 pigs; we have added the same statement in the Figure 1 legend: “Partial sampling was performed on five pigs that succumbed during the experimental process”.
- Please provide the animal ethics review and approval number, as it is currently lacking.
We have included the information in the manuscript. The same information was provided to the Vaccines editorial team for the ethics review and approval number.
- Please modify the format of the references to comply with the journal's requirements.
The references have been formatted following the MDPI guidelines.

Reviewer 4 Report
Comments and Suggestions for Authors
In this manuscript, Crisci et al. report that the challenges and lessons learned from a field trial on the understanding of porcine respiratory disease complex. They highlight the beneficial impact of high weaning weight and underscore the potential of NanoString technology for comprehensive pathogen surveillance and identify PRRSV-S and PCMV as key viral predictors of PRDC outcome. While this study is promising and generally well-designed, I have several suggestions that may help improve the manuscript.
- Page 7 of 24, ‘incubated for 2h at room temperature….. dilution for 1h at room temperature followed…..’ in the first paragraph, there should be one space between numbers and units.
- Merge 3.7.1 to 3.7.7 and summarize the results based on their correlation with of PRRSV level.
Author Response
In this manuscript, Crisci et al. report that the challenges and lessons learned from a field trial on the understanding of porcine respiratory disease complex. They highlight the beneficial impact of high weaning weight and underscore the potential of NanoString technology for comprehensive pathogen surveillance and identify PRRSV-S and PCMV as key viral predictors of PRDC outcome. While this study is promising and generally well-designed, I have several suggestions that may help improve the manuscript.
- Page 7 of 24, ‘incubated for 2h at room temperature….. dilution for 1h at room temperature followed…..’ in the first paragraph, there should be one space between numbers and units.
We have changed the text accordingly.
- Merge 3.7.1 to 3.7.7 and summarize the results based on their correlation with of PRRSV level.
We thank the reviewer for the valuable suggestion. Yet, since the presented data are rather complex, we prefer to keep the more structured approach for enhanced readability and digestibility.
